# Diabetic Complication Profiles and Associated Risk Factors: A Comprehensive Analysis from Two Public Hospitals in the Najran Region, Southern Saudi Arabia

**DOI:** 10.3390/medicina61101871

**Published:** 2025-10-18

**Authors:** Farooq Wani, Saeed AlMutyif, Altaf Bandy, Ashokkumar Thirunavukkarasu, Ekremah Alzarea, Muath Alsurur, Basil Alomair

**Affiliations:** 1Department of Pathology, College of Medicine, Jouf University, Sakaka 72388, Saudi Arabia; 2College of Medicine, Jouf University, Sakaka 72388, Saudi Arabia; saeedalmuteef@gmail.com (S.A.); muathalsurur@gmail.com (M.A.); 3Department of Clinical Sciences, College of Medicine at Shaqra, Shaqra University, Shaqra 15551, Saudi Arabia; ahbanday@su.edu.sa; 4Department of Community Medicine, College of Medicine, Jouf University, Sakaka 72388, Saudi Arabia; ashokkumar@ju.edu.sa; 5Hematopathology, Department of Pathology, College of Medicine, Jouf University, Sakaka 72388, Saudi Arabia; eaalzare@ju.edu.sa; 6Internal Medicine and Endocrinology, Department of Medicine, College of Medicine, Jouf University, Sakaka 04631, Saudi Arabia; alomairbasil@gmail.com

**Keywords:** diabetes mellitus, diabetic complications, T1D patients, T2D patients, risk factors, microvascular complications, macrovascular complications

## Abstract

*Background and Objectives:* Diabetic complications represent a major healthcare challenge globally. The Kingdom of Saudi Arabia has one of the highest prevalence rates, yet comprehensive data on complication profiles from southern regions remain limited. This study characterizes the spectrum of diabetic complications and identifies associated risk factors in the Najran region of southern Saudi Arabia. *Materials and Methods:* A hospital-based retrospective analysis of 500 diabetic patients from two major public hospitals in the Najran region was conducted using electronic medical records from January 2022 to December 2023. A systematic sampling approach was adopted. Type 1 diabetes (T1D), Type 2 diabetes (T2D), and their complications were defined using standardized criteria. Data extraction utilized a validated proforma, and analysis employed SPSS version 20. Separate analyses were conducted for T1D and T2D, with multivariable logistic regression identifying independent predictors of complications (*p* < 0.05). *Results*: The study included 200 T1D (median age 14.0 years, IQR 3.0) and 300 T2D patients (median age 23.0 years, IQR 7.0). The high proportion of T1D patients (40%) reflects the hospital’s role as a specialized pediatric and young adult diabetes referral center. Among T1D patients, 63.5% (127/200) developed complications, predominantly microvascular, whereas 50.0% (150/300) developed complications in T2D. Poor glycemic control was the strongest predictor of complications in both groups (*p* = 0.01). Rural residence significantly increased complication risk in T2D patients (*p* = 0.02). Disease duration showed differential effects; complications appeared earlier in T1D (median 6.5 years) versus T2D (median 8.2 years). *Conclusions*: This study gives the first comprehensive analysis of diabetic complications from southern Saudi Arabia, revealing distinct patterns and associated risk factors. The findings provide regional perspective on diabetic complications in Najran, highlighting the importance of early glycemic control and equitable healthcare access. The results are not intended for nationwide generalization, rather, they point to the need for region-specific diabetes management strategies.

## 1. Introduction

Diabetes mellitus is a growing global health challenge of the 21st century, with the global burden continuing to escalate at an alarming pace. According to the International Diabetes Federation (IDF) Atlas 11th Edition, approximately 589 million adults (aged 20–79 years) are living with diabetes, with the numbers predicted to rise to 853 million by 2050 [1]. This epidemic affects developed as well as developing nations, where urbanization, lifestyle changes, and genetic predisposition come together to create the perfect environment for diabetes [2]. This condition poses a significant and growing burden on global healthcare systems due to its high prevalence and associated complications [3].

The Kingdom of Saudi Arabia is experiencing one of the highest diabetes prevalence rates globally. The IDF Atlas 2024 reports that Saudi Arabia is among the top ten countries with highest prevalence, with an estimated 5.3 million adults living with diabetes [4]. The economic burden is equally staggering, with diabetes-related healthcare expenditure in Saudi Arabia reaching USD 7.3 billion in 2024, representing a significant portion of total healthcare spending [4]. The ever-increasing numbers of diabetic patients have been attributed to several factors, which includes the rapid population growth, urbanization, aging, and lifestyle changes such as unhealthy dietary habits and a sedentary style of living [5]. This trend highlights the compelling demand for the effective implementation of preventive, early diagnostic, and appropriate management strategies to ameliorate the ever-increasing disease burden.

Diabetes mellitus is broadly categorized into two major subtypes: type 1 and type 2 diabetes. T1D is an autoimmune disease with destruction of pancreatic beta cells resulting in absolute insulin deficiency, often appearing in the childhood or adolescence [6]. In Saudi Arabia, T1D incidence has shown a concerning upward trend, with the country ranking fourth in the world in terms of the incidence rate (33.5 per 100,000 individuals) of TID. Conversely, T2D arises from insulin resistance and beta-cell dysfunction, usually in adulthood [7]. The T2D epidemic in Saudi Arabia has been attributed to rapid socioeconomic development, urbanization, sedentary lifestyles, and dietary shifts toward processed foods high in refined carbohydrates [8]. While T1D and T2D share common long-term consequences of hyperglycemia, including vascular damage, their underlying pathophysiology and complication patterns differ significantly, necessitating distinct management approaches [3,9].

Diabetic complications are a major contributor to morbidity and mortality. Both subtypes of diabetes are characterized by the presence of microvascular complications, like nephropathy, retinopathy, and peripheral neuropathy, which result from prolonged exposure to hyperglycemia [10]. Macrovascular complications, including cardiovascular disease, stroke, and peripheral artery disease, are also common, particularly in T2D, due to the added impact of metabolic syndrome and advanced age [7]. Even though glycemic control is of paramount importance in diabetes treatment, other factors which play an important role include elevated blood pressure, deranged lipid metabolism, and obesity; all these factors significantly influence the risk of complications.

Recent evidence highlights important differences in the prevalence and predictors of complications between T1D and T2D. While both subtypes exhibit similar frequencies of microvascular complications, T2D patients are at a significantly increased risk of macrovascular complications due to the synergistic effects of insulin resistance, abnormal lipid levels, and chronic inflammation [10]. Conversely, T1D patients are more prone to complications linked to glycemic variability and longer disease duration [11]. Current studies have identified factors such as poor glycemic control [12], elevated HbA1c levels [13], blood pressure, and cholesterol levels as key predictors of complications [14,15,16]. However, there is a critical gap in understanding the distinct pathways and mechanisms driving these complications in T1D versus T2D, particularly in diverse populations and healthcare settings.

The Najran province, located in southern Saudi Arabia, presents unique demographic and healthcare challenges. With a population of approximately 592,300 inhabitants, the region faces distinct healthcare delivery challenges. The population is characterized by a higher proportion of young individuals, significant rural–urban disparities, and traditional lifestyle patterns that may influence diabetes management and complication development. Although several studies from central and western Saudi Arabia have examined diabetes complications [17,18], no large-scale research has been conducted in the southern region. Our study aims to fill this gap by separately analyzing complication profiles in T1D and T2D patients, providing comparative insights within the same healthcare setting.

The objective of this study was to describe the complication profiles of patients with type 1 diabetes (T1D) and type 2 diabetes (T2D) separately. We also aimed to identify demographic, clinical, and socioeconomic predictors in each group and to examine regional patterns that could guide targeted interventions and support evidence-based diabetes care planning in the Najran region.

## 2. Materials and Methods

### 2.1. Study Population and Sampling

The study population comprised T1D and T2D patients with established diagnoses attending the two public hospitals of Najran region. This hospital-based retrospective study analyzed the electronic health records of T1D or T2D patients for a period of two years from January 2022 to December 2023. Najran hospitals serve urban and rural populations with pediatric referral for T1D cases. A systematic sampling approach was used to ensure representativeness while maintaining feasibility. From an estimated 2847 total diabetic patients in the hospital database, every 6th patient was selected using computer-generated random numbers, yielding 500 patients for analysis.

### 2.2. Sample Size Calculation

The sample size calculation was performed using the formula for cross-sectional studies: n = Z^2^p(1 − p)/d^2^, where Z = 1.96 for 95% confidence interval, *p* = expected complication prevalence of 50% (based on regional estimates), and d = desired precision of 5%. This yielded a minimum required sample of 384 patients. The final sample of 500 patients provided adequate power (>90%) to detect clinically meaningful associations with a 5% significance level [19].

### 2.3. Inclusion and Exclusion Criteria

Inclusion criteria: Confirmed diagnosis of diabetes mellitus (T1D or T2D) for ≥6 months; age ≥ 5 years at time of data collection; complete medical records with ≥2 clinic visits; and residence within Najran catchment area.

Exclusion criteria: Gestational diabetes or secondary diabetes; incomplete medical records (<6 months follow-up); patients with terminal illnesses affecting diabetes management; and Type 3c diabetes or monogenic diabetes syndromes.

### 2.4. Diabetes Classification Criteria

Diabetes classification was performed according to the American Diabetes Association criteria. T1D and T2D were differentiated using standardized criteria including the following:-Age at onset;-Clinical presentation;-Autoantibody testing results;-Insulin dependency patterns.

Type 1 Diabetes (T1D): younger age at diagnosis (<35 years), absolute insulin dependence from diagnosis, presence of islet auto-antibodies, where available, clinical phenotype consistent with autoimmune diabetes, and absence of metabolic syndrome features at diagnosis.

Type 2 Diabetes (T2D): Clinical onset typically after age 25 years (or earlier with strong family history), initial response to lifestyle modifications ± oral medications, clinical phenotype consistent with insulin resistance, presence of metabolic syndrome features, and gradual progression to insulin dependence (if applicable).

### 2.5. Complication Definitions

All complications were defined using established clinical criteria, as follows:

Diabetic Retinopathy: ophthalmological examination showing microaneurysms, hemorrhages, exudates, or neovascularization consistent with diabetic eye disease [20].

Diabetic Nephropathy: persistent albuminuria (≥30 mg/g creatinine), persistently reduced eGFR (<60 mL/min/1.73 m^2^), or both for >3 months [21].

Diabetic Neuropathy: according to the criteria laid by American diabetes association [22].

Diabetic Foot Disease: presence of foot ulceration and foot infection in the setting of diabetes [23].

Macrovascular Complications: documented coronary artery disease, cerebrovascular disease, or peripheral arterial disease by appropriate imaging or clinical assessment [24].

### 2.6. Data Validation

To minimize underreporting, we implemented a multi-step validation process: (1) cross-referencing with laboratory results for diabetic nephropathy, (2) reviewing ophthalmology reports for retinopathy diagnosis, and (3) validating neuropathy diagnoses with neurological examination records. Cases with incomplete validation were excluded from complication-specific analyses.

### 2.7. Data Collection

Data extraction was performed by utilizing a standardized pre-tested proforma. Two investigators independently extracted the data, and any discrepancies were resolved through consensus with the principal investigator. Inter-rater reliability was assessed using a random sample of 50 records, yielding substantial agreement (κ = 0.87).

The proforma captured the demographic details, type of diabetes, duration of diabetes, treatment, and reported complications.

### 2.8. Statistical Analysis

SPSS software, version 20.0 for Windows (SPSS, Inc., Chicago, IL, USA), was used for analyzing the data. Descriptive analysis included frequencies for qualitative data and median and inter-quartile range (IQR) for quantitative data. Chi-square tests were used to analyze any relationship between qualitative variables. The Mann–Whitney U test was applied to compare the quantitative data. Logistic regression analysis was carried out to analyze the effect of various demographic characteristics on the T1D and T2D diabetes complications. A *p* value of <0.05 was considered significant.

### 2.9. Ethical Considerations

The study protocol was approved by the Institutional review board of King Khalid Hospital, Najran, Saudi Arabia, via order no: 2024-16A, dated, 15 September 2024. Patient confidentiality was maintained throughout the study, with all data de-identified prior to analysis. The study was conducted in accordance with the Declaration of Helsinki principles.

## 3. Results

The study population consisted of 500 diabetic patients from two public hospitals in the Najran region, including 200 (40%) with T1D and 300 (60%) with T2D. The higher proportion of T1D patients compared to the general population estimates (typically 5–10%) can be explained, since these hospitals are referral centers for regional pediatric and young adult diabetes patients, receiving cases from across the Najran province. Our analysis specifically examined T1D and T2D as distinct entities, with separate statistical analyses conducted for each type.

### 3.1. Type 1 Diabetes Patient Profile

Among the 200 T1D patients, the median age was 14.0 years (IQR 3.0, range 10–40 years), with 112 (56%) being female. The majority (156, 78%) were unemployed, reflecting the young age distribution, with most patients being students. The median disease duration was 7.2 years, with 148 (74%) having a disease duration of 6–10 years. Poor glycemic control (HbA1c > 9%) was observed in 127 (63.5%) patients, indicating significant management challenges in this population.

The geographic distribution showed 102 (51%) from urban areas and 98 (49%) from rural communities. A family history of diabetes was present in 117 (58.5%) patients, suggesting a strong genetic predisposition. All T1D patients required insulin therapy, with 72 (36%) on insulin monotherapy and 128 (64%) on insulin plus oral medications (Table 1).

### 3.2. Type 2 Diabetes Patient Profile

Among the 300 T2D patients, the median age was 23.0 years (IQR 7.0, range 13–65 years), with 137 (45.7%) being female. The employment rates were higher than in the T1D population, with 200 (66.7%) being employed. The disease duration depicted variability with 43 (14.3%) with up to 5 years, 176 (58.7%) with 6–10 years, and 81 (27%) with > 10 years.

The glycemic control was better than in the T1D population, with only 10 (3.3%) having poor control, 214 (71.3%) moderate control, and 76 (25.3%) good control. The geographic distribution showed 164 (54.7%) from urban areas. Hypertension was more prevalent (70, 23.3%) compared to the T1D patients (32, 16%). The treatment modalities included oral medications alone (145, 48.3%), insulin monotherapy (0%), and combination therapy (155, 51.7%).

The T2D patients were significantly older (median age: 23 vs. 14 years), heavier (median weight: 84 vs. 48 kg), and had high BMI (median: 29.2 vs. 25.1; *p* < 0.001) compared to the T1D patients. Poorer glycemic control (median HBA1c: 9.79 vs. 7.89; *p* < 0.001) and higher unemployment rates (78% vs. 33.3%; *p* < 0.001) were frequently observed among T1D patients. Regarding treatment modalities, T1D patients were exclusively using insulin, whereas T2D patients relied primarily on oral medications (*p* < 0.001). A longer disease duration, higher prevalence of hypertension, and a stronger family history of diabetes (*p* < 0.05 for all) was observed in T2D patients. (Table 1)

### 3.3. Complication Prevalence and Patterns

A significant association was noticed between the glycemic control and the occurrence of diabetic complication. Complications were more frequently observed in poor glycemic patients compared to moderate or good control (*p* = 0.01). Rural T2D patients exhibited more complications than their urban counterparts (*p* = 0.02). Significant associations were not observed between complications and other variables such as sex, family history of diabetes, or history of hypertension. (Table 2)

Among 200 T1D patients, 127 (63.5%) developed at least one complication. Microvascular complications were observed in 88 patients (44%) and macrovascular complications in 21 patients (10.5%). Among 300 T2D patients, 150 (50%) developed complications. Microvascular complications were seen in 136 patients (45.3%), and macrovascular complications were observed in 9 patients (3%) (Table 3).

The most common complication observed overall was neuropathy (91 cases), followed by retinopathy (83 cases) and nephropathy (50 cases). Microvascular complications were more common in T2D (e.g., retinopathy: 48 vs. 35 cases, *p* = 0.001). T1D patients had a higher prevalence of foot ulcers (13 vs. 3 cases, *p* = 0.001) and ischemic heart disease (13 vs. 5 cases), but these complications had lower rates of occurrences. A statistically significant difference in the overall distribution of complications between T1D and T2D patients was observed (χ^2^ = 22.0 and *p*-value = 0.001) (Table 3).

Graphical visualizations to illustrate the glycemic control differences between T1D and T2D (Figure 1) and the distribution of major complications across subtypes (Figure 2) were developed. These figures highlight the magnitude of differences in control and complication burden between the groups.

### 3.4. Risk Factor Analysis

Increasing age slightly decreased the odds of the occurrence of complications [OR = 0.96 (0.93–0.99); *p* = 0.04]. Patients with moderate glycemic control were significantly less likely to develop complications compared to poor glycemic control patients [OR = 0.26 (0.08–0.76); *p* = 0.01)]. The BMI, HbA1c levels, treatment modality, and diabetes duration did not show any significant predictive value for complications in this model. The overall model fit was modest, with an R^2^McF of 0.0791 (Table 4).

Model fit statistics were evaluated to assess the adequacy and explanatory power of the regression models, as summarized in Table 5.

### 3.5. Multicollinearity Assessment

We conducted comprehensive Variance Inflation Factor (VIF) analysis post hoc to validate our analytical approach (Appendix A). The key findings were as follows:Diabetes Type: VIF = 28.34 (High multicollinearity);BMI: VIF = 8.57 (Moderate multicollinearity);Treatment Modality: VIF = 7.71 (Moderate multicollinearity);Other variables: VIF < 5 (Low multicollinearity).

The high VIF for diabetes type reflects real clinical reality rather than statistical artifact. In our cohort,

-Type 1 diabetes: predominantly younger patients (mean age: 14.7 ± 4.0 years);-Type 2 diabetes: predominantly older patients (mean age: 26.0 ± 9.8 years);-Perfect treatment separation: T1D = insulin only, T2D = includes oral meds;-Distinct BMI profiles: T1D = 25.1 ± 0.7 kg/m^2^, T2D = 29.2 ± 0.4 kg/m^2.^

The multicollinearity identified in the post hoc analysis validates rather than invalidates our methodological approach. The strong associations between diabetes type and other clinical variables reflect fundamental biological and clinical distinctions between type 1 and type 2 diabetes.

We conducted sensitivity analyses to confirm our findings were robust to alternative model specifications and variable categorizations (Table 6).

### 3.6. Multiple Testing Correction

Given the multiple comparisons, the Benjamini–Hochberg False Discovery Rate (FDR) procedure was applied (Appendix A). Out of 52 *p*-values tested, 20 were initially significant (*p* < 0.05). After FDR adjustment, 16 remained significant, resulting in a net reduction of four variables. Importantly, strong associations such as age, weight, BMI, and HbA1c retained statistical significance, reflecting robust effects with very low *p*-values (often <1 × 10^−40^). Conversely, borderline associations did not retain significance under strict Bonferroni correction but remained within the False Discovery Rate (FDR) threshold. Overall, the additional analyses support the robustness of the main findings, with key predictors (age, BMI, HbA1c, weight) maintaining statistical significance after rigorous FDR correction.

## 4. Discussion

This study provides the first comprehensive analysis of diabetic complication profiles from the Najran region in southern Saudi Arabia, examining 500 patients from two major public hospitals serving as the region’s primary diabetes care centers. This study analyzed the socio-demographic characteristics, glycemic control, treatment modalities, and complications patterns among T1D and T2D patients. The findings provide valuable information regarding diabetes-related complications, offering a nuanced understanding of how these differ between diabetes types and settings. These observations underscore the usefulness of personalized approaches in diabetes management, addressing the unique needs of each patient subgroup.

### 4.1. Type 1 Diabetes Findings

Our T1D population demonstrated a high complication burden (63.5%). Several international studies have found high prevalence of complications in T1D patients [25,26,27,28]. A high prevalence of neuropathy (56%) was observed in a study conducted by Akbar DH et al., 2000, in Saudi Arabia [29]. Diabetic retinopathy is a common complication of T2D in Saudi Arabia, and the prevalence varies from 6.25% to 88.1% and is expected to increase [30]. The predominance of microvascular complications (44% of patients) reflects the characteristic pathophysiology of T1D, where absolute insulin deficiency leads to sustained hyperglycemia and subsequent vascular damage [31].

Our study reveals higher rates of macrovascular complications in T1D patients compared to international pediatric studies, possibly reflecting longer disease duration and management challenges. Evidence indicates that T1D patients may face a disproportionately high burden of macrovascular complications. Many systemic reviews and meta-analyses have demonstrated that the cardiovascular disease (CVD) risk in T1D patients is markedly elevated, with a two to tenfold increase compared to the general population [32,33,34]. Moreover, findings from the SEARCH CVD study found that T1D patients experience early onset of atherosclerosis along with clustering of cardiovascular risk factors, underscoring the rapid development of macrovascular disease [35].

The high prevalence of poor glycemic control (63.5%) in our T1D cohort is concerning but consistent with regional studies. An increased incidence of poor glycemic control has been observed in T1D patients throughout Saudi Arabia with figures ranging from 43.2 to 75.4% [36,37,38]. This pattern likely reflects multiple factors including limited access to continuous glucose monitoring, insulin pump therapy, and diabetes education resources.

### 4.2. Type 2 Diabetes Regional Patterns

The T2D complication rate (50%) in our cohort is consistent with rates reported in Middle Eastern populations (45–65%) [39]. However, the relatively young age (median 23 years) of our T2D population is noteworthy and draws attention to the concerning trend of early-onset T2D in the Gulf region [40,41]. Our findings align with recent reports from the Gulf region, where early onset of T2D and high complication rates have been documented (Al-Rubeaan et al., 2015; Aljulifi et al., 2021; Alharbi et al., 2024) [41,42,43]. These studies, together with earlier foundational work such as the UKPDS, highlight both persistent and emerging challenges in diabetes management across different settings [44]. We observed retinopathy, neuropathy, and nephropathy in 32.0%, 39.3% and 19.3% of the T2D patients. Our findings are consistent with Saudi studies such as Saiyed et al. (2022, Tabuk) and Ewid et al. (2003, Qassim), which reported high rates of microvascular complications [17,45]. Ewid et al., 2003, reported from the Qassim region of Saudi Arabia and observed retinopathy, neuropathy, and nephropathy in 42.5%, 32.5%, and 12% of T2D patients, respectively [17]. A few international studies have reported very high rates of complications in type 2. Ikem RT et al. observed microvascular complications in 69.3% and macrovascular complications in 49% of their T2D patients [46]. This may reflect regional disparities like high prevalence of diabetes mellitus and the longer duration observed in these populations. In our T2D cohort, the predominance of neuropathy over retinopathy contrasts with some international reports but aligns with recent Middle Eastern research [39,47]. This pattern may be reflective of regional diagnostic practices, healthcare seeking behavior, and genetic predispositions.

Rural residence emerged as the strongest predictor of T2D complications in our study. This finding has significant public health implications for Najran region, where approximately 30% of the population resides in rural areas. Rural T2D patients face several challenges including limited access to specialized care, dietary counseling, and inconsistent medication availability.

### 4.3. Comparisons Between T1D and T2D Patients

Significant differences were observed between the two groups of diabetic patients with respect to age, weight, BMI, and glycemic control. We observed that T2D patients were significantly older, obese or overweight, and had higher BMI’s (*p* < 0.001) when compared to the T1D patients. The findings are consistent with the previous published research and implicate the role of obesity and increasing age as strong risk factors in the etiopathogenesis of T2D. Daousi et al., 2006, also observed in their study that 86% of the T2D patients were overweight or obese, were significantly older, and had higher BMI, as well as higher prevalence of hypertension, as compared to the T1D patients [48].

T2D patients presented with longer duration of disease, a higher prevalence of hypertension, and a stronger family history (*p* < 0.05 for all), thereby augmenting the well-documented genetic as well as environmental risk factors related to T2D. Similar observations were made by Song (2015) in their cohort of T2D patients [49]. A systematic review conducted by Ismail L et al., 2021 concluded that positive family history, hypertension, obesity, and physical inactivity are strongly linked to the pathogenesis of T2D [50]. Treatment modalities differed significantly, with T1D patients predominantly using insulin, while T2D patients were managed primarily on oral hypoglycemic agents (*p* < 0.001).

Poorer glycemic control was observed in T1D patients (median: 9.79%) compared to T2D patients (median: 7.89%; *p* < 0.001). The finding that T1D patients exhibited poorer glycemic control aligns with prior studies and may be attributed to the complexity of insulin therapy and adherence challenges [51,52,53]. Additionally, higher unemployment rates in T1D patients (78% vs. 33.3%; *p* < 0.001) may lead to inadequate healthcare as well as self-management challenges. Apperley LJ and Ng SM, 2017, in their study on T1D patients, observed that poor glycemic control was notably associated with unemployment and lower educational levels [54].

The significant chi-square test result (*p* = 0.001) illustrated that the distribution of complications differed between T1D and T2D patients. The higher prevalence of retinopathy, neuropathy, and nephropathy in T2D may be ascribed to several factors, which include prolonged duration of asymptomatic hyperglycemia before diagnosis, as well as other synergistic factors like dyslipidemias, hypertension, smoking, and the duration of diabetes [55]. Higher rates of microvascular complications in T2D patients have been observed in research conducted in Saudi Arabia [17,45]. These complications emphasize the usefulness of early screening and intervention in T2D patients to limit severe disease progression.

T1D patients had a higher prevalence of foot ulcers, IHD, and stroke. The increased frequency of foot ulcers in T1D (13 vs. 3 cases) may be attributed to the longer disease duration and more frequent episodes of severe glycemic fluctuations. The relatively higher number of IHD cases (13 vs. 5) in T1D might be associated with the pro-inflammatory state, genetic polymorphisms, hypertension, etc. [35].

Given the statistically significant difference in the overall complication rate, targeted screening and prevention strategies tailored to each diabetes type must be adopted. In T2D, aggressive management of microvascular complications (e.g., nephropathy, retinopathy, and neuropathy) must be a priority, whereas in T1D, early intervention strategies for foot ulcers and cardiovascular risks should be emphasized.

### 4.4. Associations Between Complications and Variables

The study detected a significant association between glycemic control and the incidence of diabetic complications. Patients having poorer glycemic control presented with higher rates of complications (*p* = 0.01), highlighting the relevance of sustaining optimal glucose levels to reduce morbidity and mortality. Several studies have highlighted the role of poor glycemic control in increasing the diabetes related complications [56,57,58]. Internationally, the strong association we observed between poor glycemic control and complications mirrors landmark evidence from the UKPDS (Stratton et al.) [44] and DCCT (2002) [59], reinforcing the importance of sustained glycemic control. A prospective observational study by Stratton et al., 2000, observed that every 1% reduction in HbA1c reduced the rate of microvascular complications by 37% and also decreased the risk of diabetes related death by 21%, thereby signifying the role of appropriate glycemic control [44]. The DCCT/EDIC study demonstrated that intensive treatment reduces microvascular complications by 35–90% in T1D patients [59]. Our findings support this relationship, with poor glycemic control being the strongest predictor of complications in both diabetes types.

Noticeably, the rural T2D patients had increased rates of complications when compared to their urban counterparts (*p* = 0.02), which underscores the possibilities of disparities in primary and specialized healthcare access, health education, differences in lifestyle, as well as delayed diagnosis. A significant number of studies have depicted higher rates of morbidity due to diabetes-related complications in rural patients [60,61,62]. Our findings are consistent with global data showing that rural patients with diabetes are less likely to achieve optimal diagnosis, treatment, and control as well as management of related cardiovascular risk factors [63]. A study from Pakistan reported that rural T2D patients more often experienced low education, poor healthcare access, reliance on untrained practitioners, limited health awareness, lack of routine investigations, and medication non-adherence, all of which were linked to higher rates of diabetic complications [64]. Addressing these gaps through targeted interventions, such as telemedicine and diabetes education programs, is essential.

Studies from other regions, including Ndetei et al. (2024, Kenya) [64], further illustrate how demographic and contextual factors shape complication risks. The present work adds to this body of evidence by uniquely documenting regional and rural disparities in Najran, an aspect less frequently highlighted in Saudi research. In our study, significant associations were not observed between complications and other variables such as sex, family history of diabetes, or history of hypertension. The impact of sex of the patient on diabetic complications appears to be minimal, as the differences fail to reach significance. A study conducted to evaluate the impact of gender on chronic complications in T2D patients failed to find significant differences in the prevalence of complications, except for coronary artery disease, which was more prevalent in men [65]. While family history is a well-documented risk factor for developing diabetes, direct association with complications in existing diabetic patients is less clear. A recent research explored the association of family history of diabetes with diabetic complications and found that the family history did not significantly influence the development of specific complications such as nephropathy, neuropathy, cardiovascular diseases, and diabetic foot [66]. Although the presence of hypertension increased the prevalence of diabetic complications, it failed to reach statistical significance in our study. However, a strong association of hypertension with the occurrence of diabetic complications is well documented in the literature [67,68,69].

The multi-logistic regression model observed moderate glycemic control as a significant protective factor against complications (OR = 0.26; *p* = 0.01). The prevalence of diabetic complications has been associated with poorer glycemic controls in several studies published worldwide [17,51,70,71]. Unlike previous studies, BMI, HbA1c, type of treatment, and diabetes duration did not significantly predict complications in our logistic model. This may be due to limitations of sample size or other unmeasured confounders such as adherence to medication, dietary habits, or social determinants of health. The model’s overall fit (R^2^McF = 0.0791) suggests that additional factors, probably including lifestyle factors, genetic predisposition, and access to healthcare, may contribute to complication risk.

The explanatory power of our regression model was modest (R^2^McF = 0.0791), indicating that less than 8% of the variance in diabetic complications was explained by the included predictors. This suggests that important predictive variables or interactions were not captured. This highlights the complex multifactorial nature of diabetic complications and the need for more comprehensive models incorporating genetic, environmental, and behavioral factors.

Our study period coincided with the COVID-19 pandemic period, which significantly impacted diabetes care globally. Healthcare disruptions, limited access to health care, lifestyle changes, and increased psychosocial stress during lockdowns may have contributed to suboptimal glycemic control and higher complication rates, representing an important unmeasured confounder in our analysis.

This study contributes to the diabetes literature by presenting subtype-specific complication patterns, where T1D patients demonstrated earlier onset and higher risks of poor glycemic control, foot ulcers, and cardiovascular disease, while T2D patients showed a higher prevalence of microvascular complications. Rural residence emerged as a significant risk factor for complications in T2D, underscoring equity gaps in healthcare provision. Furthermore, the young median age of T2D onset (23 years) in our cohort aligns with reports from Gulf countries and highlights an urgent need for preventive strategies in younger populations.

### 4.5. Implications for Management

The findings observed in this study have many clinical and public health implications. Our results underscore the healthcare delivery challenges in the Najran region. The high proportion of T1D patients (40%) at both hospitals, compared to the typical population prevalence of 5–10%, highlight these institutions’ role as specialized referral centers for the southern Saudi Arabian region.

We recommend enhanced screening programs for microvascular complications in T2D patients and stronger glycemic control programs for T1D patients. The significant association between poor glycemic control and the increased rate of complications emphasizes the need for proper patient education and strict adherence to the treatment plans. Emphasis must be placed on early interventions, including lifestyle modifications and regular monitoring, to improve patient outcomes. Furthermore, there is a need for public health strategies like healthcare provider training, telemedicine consultations, and mobile clinics to improve rural healthcare access. Community-based interventions are recommended to bridge the healthcare gaps and reduce the complication rates.

### 4.6. Limitations and Future Directions

Despite its strengths, the study has certain limitations. Firstly, our retrospective design provides valuable baseline data for the Najran region; however, we acknowledge that longitudinal follow-up studies are needed to establish causal relationships and track complication progression over time. We are currently planning a prospective cohort study to address these important questions. Second, reliance on electronic health records may introduce biases due to missing or incomplete data or may lead to underestimation of complications. There is a potential for the underreporting of complications, which we have tried to minimize by implementing a multi-step validation process. Thirdly, important confounding factors such as socioeconomic status, diet, physical activity, lipid levels, and healthcare access were not captured. These factors likely contribute to the rural–urban disparities observed and represent priority areas for future comprehensive studies in the region. Fourthly, our hospital-based sample likely represents patients with more established diabetes and potentially higher complication rates compared to the general diabetic population. Future community-based studies incorporating primary care centers and population screening would provide more generalizable prevalence estimates for the Najran region. Lastly, our findings are specific to the Najran region’s hospital-attending diabetic population and should not be extrapolated to the broader Saudi Arabian population without additional multi-regional validation studies.

## 5. Conclusions

This study offers the first comprehensive analysis of diabetic complications in the Najran region, highlighting strikingly distinct patterns between T1D and T2D. Significant differences in demographic features, glycemic control, and complications between T1D and T2D patients underscore the multifactorial nature of diabetes complications. Higher rates of obesity, longer disease duration, and higher hypertension prevalence was observed in T2D, whereas poor glycemic control was more prevalent in T1D patients.

Poor glycemic control emerged as the most consistent predictor of complications across both subtypes, while rural residence significantly magnified the risks, especially in patients with T2D. Distinct complications patterns, with T2D patients showing higher microvascular complications and T1D patients at increased risk of foot ulcers and cardiovascular issues, were observed. The earlier development of complications in T1D and the disproportionate burden of microvascular disease in T2D emphasize the need for subtype-specific and evidence-driven management strategies. Our findings highlight poor glycemic control and rural residence as key risk factors for complications in Najran. While approaches such as telemedicine and mobile clinics may help address these challenges, their feasibility will depend on regional healthcare resources and infrastructure. Subtype-specific management strategies are warranted but should be validated in larger multicenter studies before broad implementation.

## Figures and Tables

**Figure 1 medicina-61-01871-f001:**
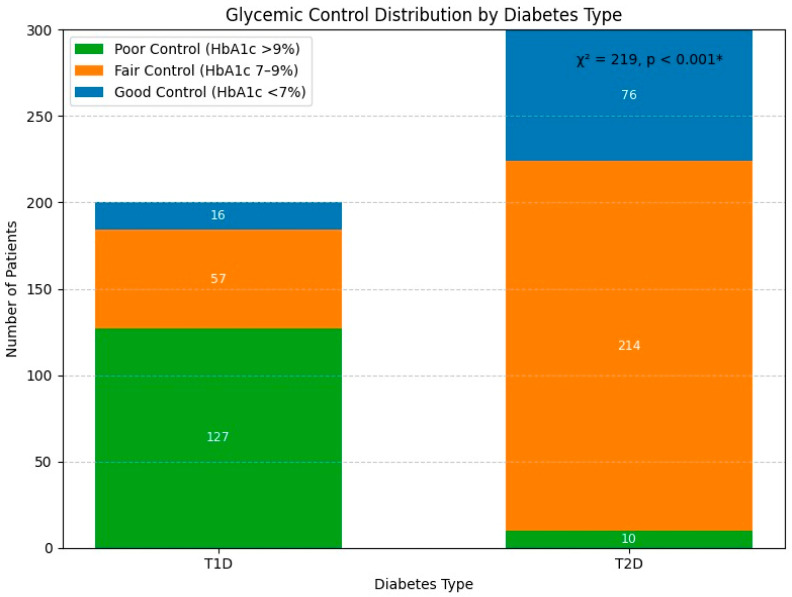
Glycemic control distribution by diabetes type.

**Figure 2 medicina-61-01871-f002:**
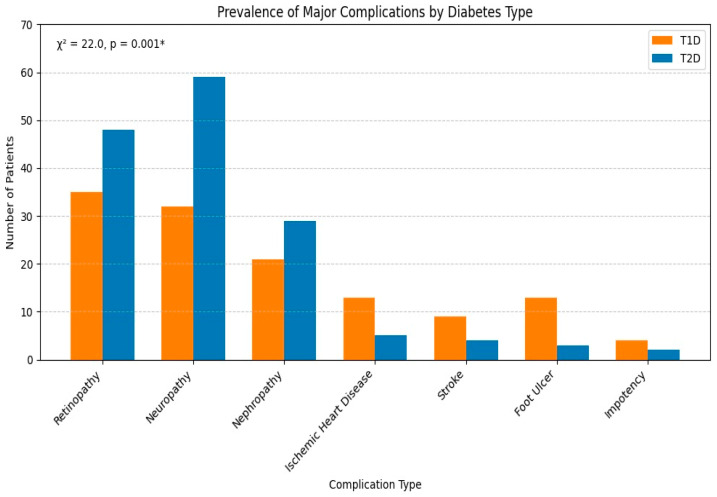
Prevalence of major complications by diabetes type.

**Table 1 medicina-61-01871-t001:** Comparison of socio-demographic characteristics between T1D and T2D patients (n = 500).

Characteristic	Type 1 Diabetes (n = 200)	Type 2 Diabetes (n = 300)	χ^2^, *p*-Value
**Gender**			
Male	88 (44.0%)	163 (54.3%)	χ^2^ = 5.13, *p* = 0.024
Female	112 (56.0%)	137 (45.7%)	
**Age (years)**			
Median (IQR)	14.0 (3.0)	23.0 (7.0)	*p* = 0.00
Range	10–40	13–65	
**Weight (kg)**			
Median (IQR)	48.0 (13.0)	84.0 (16.3)	*p* = 0.00 ^#^
Range	30–89	43–104	
**BMI (kg/m^2^)**			
Median (IQR)	25.1 (0.92)	29.2 (0.47)	*p* = 0.00 ^#^
Range	22.8–26.5	28.1–30.6	
**HbA1c (%)**			
Median (IQR)	9.79 (0.54)	7.89 (0.92)	*p* = 0.00 ^#^
Range	6.8–11.9	5.3–9.42	
**Geographic Distribution**			
Urban residence	102 (51.0%)	164 (54.7%)	χ^2^ = 0.648, *p* = 0.421
Rural residence	98 (49.0%)	136 (45.3%)	
**Employment Status**			
Employed	44 (22.0%)	200 (66.7%)	χ^2^ = 95.8, *p* < 0.001 *
Unemployed	156 (78.0%)	100 (33.3%)	
**Disease Duration**			
≤5 years	52 (26.0%)	43 (14.3%)	χ^2^ = 67.0, *p* < 0.001 *^$^
6–10 years	148 (74.0%)	176 (58.7%)	
>10 years	0 (0%)	81 (27.0%)	
**Treatment Modality**			
Insulin alone	72 (36.0%)	0 (0%)	χ^2^ = 207. 9, *p* < 0.001 *^$^
Insulin + oral agents	128 (64.0%)	155 (51.7%)	
Oral agents alone	0 (0%)	145 (48.3%)	
**Hypertension**			
Yes	32 (16.0%)	70 (23.3%)	χ^2^ = 3.97, *p* = 0.046 *
No	168 (84.0%)	230 (76.7%)	
**Family history of diabetes**			
Yes	117 (58.5%)	130 (43.3%)	χ^2^ = 11.0, *p* < 0.001 *
No	83 (41.5%)	170 (56.7%)	
**Glycemic Control**			
Poor (HbA1c > 9%)	127 (63.5%)	10 (3.3%)	χ^2^ = 219, *p* < 0.001 *
Moderate (HbA1c 7–9%)	57 (28.5%)	214 (71.3%)	
Good (HbA1c < 7%)	16 (8.0%)	76 (25.3%)	

* Significant. ^$^ Fishers exact test. ^#^ Mann–Whitney U.

**Table 2 medicina-61-01871-t002:** Comparison of complications between T1D and T2D patients (n = 500).

Diabetes Type	Characteristic	Complications		Total	χ^2^	*p*-Value
		No	Yes			
T1D and T2D	**Gender**					
	Male	114	137	251	0.13	0.71
	Female	109	140	249		
T1D	**Gender**					
	Male	36	52	88	1.31	0.25
	Female	37	75	112		
T2D	Male	78	85	163	0.65	0.41
	Female	72	65	137		
T1D	**Residence**					
	Urban	32	70	102	2.36	0.12
	Rural	41	57	98		
T2D	Urban	92	72	164	5.38	0.02*
	Rural	58	78	136		
T1D	**Duration of diabetes**					
	≤5 years	19	33	52	NA	NA
	6–10 years	54	94	148		
	>10 years	0	0	0		
T2D	≤5 years	23	20	43	0.54	0.76
	6–10 years	89	87	176		
	>10 years	38	43	81		
T1D	**Family history of diabetes**					
	No	24	59	83	3.52	0.06
	Yes	49	68	117		
T2D	No	81	89	170	0.86	0.35
	Yes	69	61	130		
T1D	**History of hypertension**					
	No	64	104	168	1.15	0.56
	Yes	9	23	32		
T2D	No	119	111	230	1.19	0.27
	Yes	31	39	70		
T1D & T2D	**Glycemic control**					
	Poor	47	90	137	8.85	0.01 *
	Moderate	135	136	271		
	Good	41	51	92		
T1D	**Glycemic control**					
	Poor	45	82	127	0.61	0.73
	Moderate	23	34	57		
	Good	5	11	16		
T2D	Poor	2	8	10	4.27	0.11
	Moderate	112	102	214		
	Good	36	40	76		

* Significant. NA: Not applicable (χ^2^ test not performed due to zero counts in this category).

**Table 3 medicina-61-01871-t003:** Specific complication types and patterns (n = 277 patients with complications).

Complication Type	Type 1 Diabetes n = 127 Patients	Type 2 Diabetes n = 150 Patients	Total n = 277	*χ^2^, p*-Value
**Microvascular Complications**				
Retinopathy	35 (27.6%)	48 (32.0%)	83 (30.0%)	
Neuropathy	32 (25.2%)	59 (39.3%)	91 (32.9%)	χ^2^ = 22.0
Nephropathy	21 (16.5%)	29 (19.3%)	50 (18.1%)	p = 0.001 *
**Microvascular Complications**				
Ischemic heart disease	13 (10.2%)	5 (3.3%)	18 (6.5%)	
Stroke	9 (7.1%)	4 (2.7%)	13 (4.7%)	
**Other Complications**				
Foot ulcer	13 (10.2%)	3 (2.0%)	16 (5.8%)	
Impotency	4 (3.1%)	2 (1.3%)	6 (2.2%)	
**Total**	**127**	**150**	**277**	

* Significant.

**Table 4 medicina-61-01871-t004:** Multi-logistic regression of the predictors of complications.

Predictor	Estimate	SE	Z	*p*	Odds Ratio (95% CI)
Intercept	12.66	9.02	1.40	0.16	3.17 × 10^−6^ (6.61 × 10^−14^–151.824)
Age	−0.03	0.017	2.00	0.04 *	0.96 (0.93−0.99)
Weight	−0.00	0.10	0.14	0.88	0.98 (0.97−1.02)
BMI	0.45	0.34	1.31	0.19	1.56 (0.79−3.07)
HBA1c	0.35	0.22	1.54	0.12	1.42 (0.90−2.23)
Sex (Female–Male)	0.08	0.27	0.30	0.75	1.09 (0.90−2.23)
Residence (Rural–Urban)	−0.04	0.29	0.15	087	0.95 (0.36−1.30)
Occupation (Employed–Unemployed)	−0.36	0.32	1.13	0.25	1.69 (0.72−2.50)
Diabetes type (Type 2–Type 1)	−0.54	1.91	0.28	0.77	0.58 (0.01−24.72)
Duration (6–10 years–Up to 5 years)	0.28	0.34	0.81	0.41	1.32 (0.67-2.60)
Duration (>10 years–Up to 5 years)	−0.02	0.62	−0.03	0.97	0.98 (0.28−3.31)
Treatment (Insulin plus OHG–Insulin)	−1.17	0.62	−1.87	0.06	0.31 (0.09−1.05)
Treatment (OHG–Insulin)	−0.36	1.18	−0.30	0.75	0.69 (0.06−7.03)
F/H Diabetes (Yes–No)	−0.31	0.28	−1.12	0.26	0.72 (0.41−1.26)
Hypertension (Yes–No)	0.41	0.37	1.12	0.29	1.15 (0.73−3.12)
Glycemic control (Moderate–poor)	−1.34	0.54	−2.45	0.01 *	0.26 (0.08−0.76)
Glycemic control (Good–poor)	−0.41	0.70	−0.58	0.56	0.66 (0.16−2.64)

* Statistically significant at *p* < 0.05.

**Table 5 medicina-61-01871-t005:** Model fit measures.

Model	Deviance	AIC	R^2^_McF_
1	314	348	0.0791

**Table 6 medicina-61-01871-t006:** Sensitivity Analyses Assessing the Robustness of Findings.

Analysis	Chi^2^	*p*-Value	Interpretation
Alternative HbA1c categories (<7, 7–8.5, >8.5)	5.48	0.0645	Association remains significant
Rural vs. Urban complications (T1D)	1.93	0.1646	No rural–urban difference in T1D
Rural vs. Urban complications (T2D)	4.86	0.0276	Rural disadvantage persists in T2D *

* Remains significant despite alternative analytical approaches.

## Data Availability

The data can be made available on request.

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
