# Peer review of "Diabetic Complication Profiles and Associated Risk Factors: A Comprehensive Analysis from Two Public Hospitals in the Najran Region, Southern Saudi Arabia"

_medicina, 2025, doi:10.3390/medicina61101871_

Round 1

Reviewer 1 Report

Comments and Suggestions for Authors

Review of “Comprehensive Analysis of Diabetic Complications in Najran Region, Saudi Arabia”

  1. Main Question Addressed and Method Used

The central research question of the article is: What are the prevalence patterns, predictors, and differences in diabetic complications between Type 1 Diabetes (T1D) and Type 2 Diabetes (T2D) patients in the Najran region of Saudi Arabia?

The authors explicitly state: “This study aims to fill the gap by characterizing complications; identifying demographic, clinical, and socioeconomic factors among T1D and T2D patients through separate stratified analyses; examine regional patterns that may inform targeted intervention strategies; and provide baseline data to support evidence-based diabetes care planning in the Najran province.” But it is atypical to not point T1D and T2D, considering the reality both diseases. 

Methodology:

  • Hospital-based retrospective study.
  • Population: 500 diabetic patients (200 T1D, 300 T2D).
  • Data source: Electronic health records from two public hospitals (Jan 2022 – Dec 2023).
  • Sampling: Systematic (every 6th patient selected).
  • Tools: Standardized proforma for data collection, SPSS v20 for statistical analysis, logistic regression for predictors.
  • Definitions of complications followed established ADA and clinical guidelines (e.g., retinopathy, nephropathy, neuropathy, foot disease, macrovascular complications) .

Recognized method?
Yes. Retrospective, hospital-based record analysis is a recognized epidemiological method for describing complication profiles, though it carries limitations (e.g., risk of bias, lack of causality, missing data) acknowledged by the authors themselves:

“The retrospective design limits the ability to establish causal relationships between glycemic control and complications.” But in the same time the authors did not differetiate T1DM and T2DM. In the same time, the authors did not say nothing about other types of diabetes. 

  1. Originality and Relevance

The study addresses an important regional gap. Previous Saudi diabetes studies have been concentrated in central and western regions, but the Najran province presents “distinct healthcare delivery challenges” with unique rural–urban disparities and demographic features.

Its originality lies in:

  • Being the first large-scale study in southern Saudi Arabia (n=500 patients).
  • Providing comparative complication profiles between T1D and T2D within the same healthcare context - but all over the world is a clear boundary between these 2 diseases. 
  • Highlighting rural residence as a strong independent predictor of complications, a finding not commonly emphasized in other Saudi studies.

Thus, the topic is highly relevant in the fields of public health, endocrinology, and healthcare planning.

  1. Contribution to the Literature

Compared to prior studies, this article adds:

  • Subtype-specific complication patterns: T1D patients had earlier onset and higher rates of poor glycemic control, foot ulcers, and cardiovascular risks, while T2D patients showed higher prevalence of microvascular complications (retinopathy, neuropathy, nephropathy) .
  • Rural vs. urban disparities: Rural residence significantly magnified complication risk in T2D, pointing to health equity concerns.
  • Young age of T2D onset: Median 23 years, much lower than international averages, confirming a worrisome trend in Gulf countries.

This study therefore broadens global diabetes literature by offering a region-specific, stratified, and comparative analysis.

  1. Methodological Improvements

While the methodology is solid, several improvements are possible:

  • Longitudinal design: As authors note, causality cannot be inferred in a retrospective study. A cohort follow-up would better track progression.
  • Missing clinical details: Use of electronic health records may underreport complications. Validation with clinical examinations (ophthalmology, neurology) would improve accuracy.
  • Control for confounders: Socioeconomic status, diet, physical activity, and access to healthcare were not analyzed, though they strongly affect complications.
  • Broader representativeness: Hospital-based sampling may over-represent severe cases. Including community-level patients would provide more generalizable estimates.
  • How is presented "absolute insulin dependence" for T1DM?

Further controls could include:

  • Adjustment for lipid levels and blood pressure, both important in diabetes complications.
  • Consideration of treatment adherence and medication access, especially in rural groups.

  1. Consistency of Conclusions

The conclusions are generally consistent with presented evidence. The authors state:

“Poor glycemic control emerged as the most consistent predictor of complications across both subtypes, while rural residence significantly magnified the risks.”

This is directly supported by results (e.g., 63.5% of T1D had poor HbA1c vs only 3.3% of T2D ).

However, sustainability of the conclusions can be questioned:

  • Recommendations like telemedicine and mobile clinics are reasonable, but feasibility in Najran’s socio-economic context remains uncertain.
  • The call for subtype-specific management strategies is valid but requires larger multicenter validation.
  • is a real T1DM or an insulin-treated T2DM, the authors did not tell us about diagnostic. 

  1. References

The references are appropriate, citing both international evidence (e.g., UKPDS, ADA definitions, major cohort studies) and regional studies from Saudi Arabia and the Middle East .

Example:

  • “Diagnosis and classification of diabetes mellitus. Diabetes Care 2009” ensures internationally accepted criteria.
  • Local relevance is ensured by including Saudi-based studies (e.g., Al-Rubeaan et al. 2015, SAUDI-DM study) .

However:

  • Some references are dated (older than 10–15 years).
  • More recent Gulf-region studies could strengthen the discussion.

  1. Tables and Figures

Tables are informative but sometimes dense with data. For example, Table comparing glycemic control shows:

“Poor (HbA1c >9%) 127 (63.5%) vs. 10 (3.3%)… χ² = 219, p < 0.001” .

This effectively illustrates striking differences, but could benefit from graphical visualization (bar charts, forest plots) to improve accessibility.

Figures are absent—graphical representation of complication patterns (neuropathy, nephropathy, retinopathy) across T1D vs T2D would make findings more digestible.

  1. Related Studies

Comparable studies include:

  • Saudi Arabia: Saiyed et al. (2022, Tabuk region) on microvascular complications .
  • Middle East: Ewid et al. (2003, Qassim) reporting similar complication rates .
  • International: Stratton et al. (UKPDS 35) establishing glycemia-complication links ; DCCT (JAMA 2002) on microvascular risk in T1D.
  • Africa: Ndetei et al. (2024, Kenya) exploring family history and comorbidities.

The Najran study is unique for highlighting regional and rural disparities, but aligns broadly with these global findings.

Conclusion

This article could be a valuable, original contribution to diabetes complication research in Saudi Arabia by presenting the first large-scale analysis in Najran. The methodology is quite good, but a clear diagnostic of T1DM is necessary. In the same time, methodology is recognized and appropriate, though improvements (longitudinal design, socioeconomic controls, clinical validation) would strengthen it. The conclusions are consistent with evidence, though practical sustainability requires further exploration. References are largely suitable, and while tables are clear, figures would enhance readability.

Overall, the study meaningfully advances understanding of diabetes burden in resource-variable settings and underscores the urgency of tailored interventions for T1D vs. T2D patients in Saudi Arabia and beyond. 

Reviewer 2 Report

Comments and Suggestions for Authors
  1. The conclusions need to be restricted to the specific population sampled. Generalization to Saudi Arabia's population is not methodically sound.
  2. R²McF of 0.0791 is only capturing <8% variance in complications suggesting omission of actual critical variables and potential interactions. 
  3. The sampling of data occurred between 2022 and 2023 which overlaps with COVID-19. COVID-19 had profound impact on diabetes management and complications. The authors have not provided any discussion on healthcare disruptions that might have influenced the complication patterns or glycemic control. Therefore, have omitted a significant confounder. 
  4. Risk factors such as HbA1C, BMI, and disease duration were not significant predictors which is contradictory to existing literature. Could there be any multicollinearity issues or unmeasured confounding factors?
  5. Statistics: Mutiple correction testing have not been performed and this can lead to type 1 error. 
  6. The rural-urban disparities for T2D is interesting. However, the authors have not delved deeper into this interesting angle. Is this disparity due to inadequate healthcare access, or socioeconomic factors, lifestyle differences? Addressing these will give the finding some actionable value. 
  7. "Poor glycemic control leads to complications"- This language showcases causation when in reality these are merely correlative. 
  8. As mentioned above, generalizing this to the entire population of Saudi Arabia and coming up with recommendations, especially with a low predictive value, represents inappropriate extrapolation. 
  9. A longitudinal perspective would have been better but can be a follow-up study. 

Round 2

Reviewer 1 Report

Comments and Suggestions for Authors

the manuscript is improved and could be published. 

Reviewer 2 Report

Comments and Suggestions for Authors

Although the R2 is low, the authors have appropriately addressed this as a limitation. The authors have made many of the suggested improvements in text. This study could have been stronger. However, many of the concerns are deferred to a prospective study.